# Validation of Aerobic Capacity (VO2max) and Pulse Oximetry in Wearable Technology

**DOI:** 10.3390/s25010275

**Published:** 2025-01-06

**Authors:** Bryson Carrier, Sofia Marten Chaves, James W. Navalta

**Affiliations:** School of Integrated Health Sciences, Department of Kinesiology and Nutrition Sciences, University of Nevada, Las Vegas, NV 89154, USA

**Keywords:** cardiorespiratory fitness, fitness tracker, activity monitor, biometric technology, altitude, hypoxia

## Abstract

Introduction: As wearable technology becomes increasingly popular and sophisticated, independent validation is needed to determine its accuracy and potential applications. Therefore, the purpose of this study was to evaluate the accuracy (validity) of VO2max estimates and blood oxygen saturation measured via pulse oximetry using the Garmin fēnix 6 with a general population participant pool. Methods: We recruited apparently healthy individuals (both active and sedentary) for VO2max (n = 19) and pulse oximetry testing (n = 22). VO2max was assessed through a graded exercise test and an outdoor run, comparing results from the Garmin fēnix 6 to a criterion measurement obtained from a metabolic system. Pulse oximetry involved comparing fēnix 6 readings under normoxic and hypoxic conditions against a medical-grade pulse oximeter. Data analysis included descriptive statistics, error analysis, correlation analysis, equivalence testing, and bias assessment, with the validation criteria set at a concordance correlation coefficient (CCC) > 0.7 and a mean absolute percentage error (MAPE) < 10%. Results: The Garmin fēnix 6 provided accurate VO2max estimates, closely aligning with the 15 s and 30 s averaged laboratory data (MAPE for 30 s avg = 7.05%; Lin’s concordance correlation coefficient for 30 s avg = 0.73). However, it failed to accurately measure blood oxygen saturation (BOS) under any condition or combined analysis (MAPE for combined conditions BOS = 4.29%; Lin’s concordance correlation coefficient for combined conditions BOS = 0.10). Conclusion: While the Garmin fēnix 6 shows promise for estimating the VO2max, reflecting its utility for both individuals and researchers, it falls short in accurately measuring BOS, limiting its application for monitoring acclimatization and managing pulmonary diseases. This research underscores the importance of validating wearable technology to leverage its full potential in enhancing personal health and advancing public health research.

## 1. Introduction

Wearable technology (WT) has continued to grow in popularity and sophistication each year, with WT reaching the number one spot in worldwide surveys of fitness trends in seven of the last nine years and being in the top three for the other two years (2018 and 2021) [1,2,3,4,5,6,7,8,9]. According to recent surveys, almost one in three Americans use a wearable device to track their health and exercise, and around 70% of people own at least one wearable or plan to buy one in the next year [10,11]. This prevalence of WT may represent a revolutionary change in physiology and public health research simply due to the vast pool of potential data that may become available to researchers. Also, an important aspect is the constant monitoring of physiological metrics that these devices perform, which will provide granular details into a person’s physiology that could transform human physiology research [12,13]. However, this transformation may only come to be realized if WT devices are found to be accurate in their measurements and estimates. As these consumer-grade wearable devices are not subject to any type of regulation, there is no governing body ensuring their accuracy. Thus, if researchers, athletes/coaches, public health officials, and healthcare professionals hope to continue to utilize these devices, an understanding of their accuracy and when they can appropriately be used is necessary. This underpins the importance of independent validation of WT devices by researchers to further several scientific fields.

Among the many variables that WT can estimate or measure, the maximal aerobic capacity (or VO2max) and blood oxygen saturation (BOS) measured via pulse oximetry are important for a variety of health- and fitness-related purposes. VO2max represents the maximal amount of oxygen an individual can transport from the environment into their lungs, diffuse into the blood, and extract at the muscles and organs to produce energy, or ATP. It represents a measure of cardiorespiratory fitness (CRF) and has a strong inverse relation with all-cause mortality and cardiovascular diseases [14,15,16]. VO2max also has an important relationship to endurance performance among athletes, often being cited as the most important single factor—or among the most important factors—in predicting race performance [17,18,19]. Pulse oximeters can non-invasively measure the amount of oxygen bound to hemoglobin based on how light reflects off the blood cells when broadcast from the device. Devices with pulse oximeters to measure BOS can also be used to monitor cardiorespiratory functions, especially in people with pulmonary diseases. It can also be useful for athletes looking to travel to altitude for an event or competition who wish to monitor their acclimatization process [20,21]. Therefore, the purpose of this study was to evaluate the accuracy (validity) of VO2max estimates and blood oxygen saturation measured via pulse oximetry using the Garmin fēnix 6 with a general population participant pool.

## 2. Materials and Methods

Prior to data collection occurring for this study, the protocols were approved by the University of Nevada, Las Vegas Institutional Review Board (IRB). All participants signed an informed consent and filled out pre-assessment documents prior to completing the study. While the VO2max and pulse oximetry testing were completed separately, some participants completed both and are included in each dataset. As the participant pool for both VO2max and pulse oximetry testing are different, demographic data are provided for each group.

### 2.1. VO2max Testing

For VO2max testing, 19 apparently healthy (people who, based on their personal knowledge, reported being healthy at the time the study was conducted), active and sedentary individuals were recruited to participate (25.50 ± 5.26 years, 11 male, 8 female, 173.63 ± 9.08 cm, 74.08 ± 14.16 kg, BMI = 24.42 ± 3.21 kg/m^2^, 22.14 ± 6.06% fat mass, 36.87 ± 4.58% muscle mass, 25.07 ± 23.65 km run per week, and all reported as mean ± SD). Data collection occurred over two separate days. On the first day, participants completed a graded exercise test utilizing progressive increases in speed and grade to determine their VO2maxs. Maximal oxygen consumption was measured using the ParvoMedics TrueOne 2400 metabolic cart (ParvoMedics Inc., Salt Lake City, UT, USA). The VO2max was determined by taking the highest average oxygen consumption during the graded exercise test for a set timeframe. Aggregated VO2max values for the 4-breath, 15-s, 30-s, and 1-min averaged timeframes were obtained by the metabolic cart and served as the criteria measures for the comparisons to the WT device. The second day consisted of an outdoor run that was guided by the wearable device (Garmin fēnix 6^®^, Garmin Ltd., Olathe, KS, USA) to generate an estimated VO2max value. The Garmin fēnix 6 is a rugged, multisport GPS smartwatch designed for outdoor use and athletes. It is marketed to combine the functionality of a fitness tracker, outdoor navigator, and smartwatch in a durable, wrist-worn device. The participants were asked to come back between two and seven days from the first visit (5.06 ± 3.96 days). The researchers performed a factory reset on the watch prior to each subject to prevent data from previous participants from influencing the measurements and estimates of the current subject. The participants then put on the associated heart rate monitor (Garmin HRM-Run^®^) for the outdoor run. The outdoor run involved a 10–15 min run at an intensity above 70% of the participant’s estimated max HR, according to the manufacturer’s guidelines. This provided the device with enough data to estimate the VO2max, using a linear extrapolation of the heart rate (HR) and running speed [22]. The outdoor run was performed in one of two places: the University track or a flat area of the campus, depending on logistics and track availability. Five participants completed the testing at the track, and fourteen participants completed the testing on campus. The altitude was ~686 m, and the average temperature during outdoor testing was 20.67 ± 12.62 °C, as measured by local weather readings. The average distance, time, pace, and HR were 2.13 ± 0.17 km, 12.91 ± 1.42 min, 6.33 ± 1.49 min/km, and 153.50 ± 11.45 bpm, respectively, as measured by the device. The data collection took place over the timespan of ~14 months, with running trials being completed during the morning, afternoon, and evening.

### 2.2. Pulse Oximetry Testing

For pulse oximetry testing, 22 apparently healthy individuals were recruited to participate (25.48 ± 6.02 years, 13 male, 9 female, 173.27 ± 7.70 cm, 68.88 ± 9.10 kg, BMI = 22.91 ± 2.40 kg/m^2^, 18.55 ± 7.05% fat mass, and 38.73 ± 3.61% muscle mass). The participants began by putting on the fēnix 6 on their left wrist and were instructed to have the strap tension secure but comfortable. The researchers then placed a medical-grade pulse oximeter (Roscoe Medical Fingertip Pulse Oximeter, Model: POX-ROS, Roscoe Medical Inc., Middleburg Heights, OH, USA) on the right index finger of the participant. The participants completed eight trials of testing under four conditions (two per condition). The first testing condition was under normoxic (normal oxygen concentration) conditions, with the watch head placed on the posterior wrist. The researchers performed the necessary steps (selecting the correct icon in the watch) on the watch to generate a BOS level by the fēnix 6 and recorded the value from the fingertip oximeter at the same time the watch generated a value. Afterward, the watch was then placed on the anterior wrist, and the process was repeated. After both normoxic conditions were completed, the participants performed hypoxic (low oxygen concentration) testing of the pulse oximeter. The participants were connected to an altitude simulator machine (Hypoxico Everest Summit II, Hypoxico Inc., New York, NY, USA) for a minimum of five minutes to allow for their blood oxygen levels to stabilize prior to testing. The machine was set to an altitude of 3657.6 m (12,000 ft) as the default for participants. However, if the participants became lightheaded or uncomfortable at that simulated altitude, it was lowered to an altitude better tolerated by the individual, and a five-minute waiting period reset occurred, with the possibility of returning to normoxia for as long as needed before restarting at a lower simulated altitude. All participants were seated for all pulse oximetry tests. The participants were instructed to control their breathing rate and breathed in and out in synchronization with the altitude simulator bursts of air. This corresponded to a breathing rate of 12.5 breaths per minute. Blood oxygen saturation testing under hypoxia was tested with the watch on the anterior and posterior left wrists, as was performed prior in the normoxic testing condition. The average time under hypoxia was 9.18 ± 1.05 min. If the fēnix 6 was unable to generate a measurement of BOS for any trial, the researchers retried up to three times for each trial when the watch did not generate a value on the first attempt. If it was still unable to generate a measurement after three tries, no further attempts were made. Once the values were obtained from the watch and the fingertip oximeter, the pulse oximetry testing was concluded.

### 2.3. Data Analysis

The VO2max values for each timeframe (4-breath, 15 s, 30 s, and 1 min) and BOS values for each condition (anterior/posterior placement, normoxia/hypoxia) were input into Google Sheets (Alphabet Inc., Mountain View, CA, USA). The pulse oximetry values were compared by condition as well as the combined dataset. All granular calculations were completed within Google Sheets. All summary statistics, validation measures, and figures were completed and generated in jamovi (jamovi Project, version 2.6.19, https://www.jamovi.org/). Descriptive statistics, error analysis (mean absolute percentage error), correlation analysis (Pearson’s r, Lin’s concordance correlation coefficient [CCC]), equivalence testing (TOST paired samples test), and bias assessment (Bland–Altman analysis) were also performed. The TOST test upper and lower bounds were set at +0.5 and −0.5 Cohen’s D for each test. Data analysis for the VO2max data was completed by comparing the fēnix 6 estimates of the VO2max to each laboratory aggregated timeframe (4 breath, 15 s, 30 s, 1 min). Determination of validation was predetermined, and any device that produced a CCC > 0.7 and a MAPE < 10% was considered valid.

## 3. Results

### 3.1. VO2max

The 19 participants used for this analysis had an average VO2max of 48.9 mL/kg/min and an average VO2max percentile of 83.37 ± 21.14%, based on the 30 s averaged VO2max values. The error analysis showed that the fēnix 6 VO2max estimate had a MAPE of less than 10% for the 15 s, 30 s, and 1 min averaged timeframes (see Table 1). The correlation analysis produced a CCC > 0.7 for both the 15 s and 30 s averaged timeframes (see Table 1). Equivalence testing via the TOST test produced no equivalent results, with the equivalence conditions being violated for the 4-breath, 15 s, 30 s, and 1 min averaged times (see Table 1). The Bland–Altman bias values and 95% confidence intervals can be found in Table 1, and the associated plots can be found for all time parameters in Figure 1.

### 3.2. Pulse Oximetry

The error analysis showed that the fēnix 6 BOS values had a MAPE of less than 10% for all four conditions and the combined data (anterior/posterior, hypoxia/normoxia; see Appendix A and Appendix A). The correlation analysis did not produce a CCC > 0.7 for any conditions, including the combined data (see Appendix A and Appendix A). Equivalence testing via the TOST test was violated for all four conditions but was met for the combined data (see Appendix A and Appendix A). The Bland–Altman bias values and 95% confidence intervals can be found in Table 2 for the combined data and the Appendix A for individual conditions. The associated plots can be found for the combined data in Figure 2 The total number of measurements that the fēnix 6 generated was 52, for a total success rate (or data availability rate) of 59%. This means that when prompted for a blood oxygen saturation measurement, it only provided data 59% of the time.

## 4. Discussion

In this study, the validity of the VO2max estimates and blood oxygen saturation (BOS) values measured via pulse oximetry in wearable technology (WT) was compared to gold-standard measurements. Based on the pre-established validation criteria, the fēnix 6 has acceptable accuracy (MAPE < 10%, CCC > 0.7) in its estimation of VO2max and corresponds closely to the 15 s and 30 s averaged timeframes. The measurements of BOS via pulse oximetry did not have acceptable accuracy for any condition or the combined data. As the appropriate use cases of these devices are discussed, it is important to note that these are consumer-grade devices, not medical devices. Thus, they are not subject to FDA regulation (or any other governing body) in terms of accuracy and effectiveness. VO2max and pulse oximeters have an important role in monitoring the health of an individual, including general health and fitness levels and those with potential cardiovascular disease (CVD) and pulmonary diseases. While these devices are being used for measuring variables in diseased populations, they are not intended for that purpose. Despite this, researchers, healthcare professionals, and public health officials are utilizing WT to track these metrics for scientific-, policy-, and healthcare-related purposes [23,24,25,26,27,28,29]. This illustrates the need for an independent evaluation of these devices in terms of their validity and reliability compared to gold-standard measurements. Wearable technology has the potential to revolutionize public health and physiology research due to its constant monitoring and widespread availability [12,13]. Thus, researchers, healthcare professionals, public health officials, and scientific journals should be invested in the independent validation of these devices to further several scientific fields.

Wearable technology can generate an estimate of the VO2max through the HR, as the linear relationship between the HR and VO2 is well-established [22]. The fēnix 6 measures the users HR and running speed and utilizes a linear extrapolation up to the estimated max HR, based on an individual’s age, to determine the VO2max. While this can be accomplished simply with the watch and built-in photoplethysmography (PPG)-based HR monitor, an accessory HR monitor that is placed on the chest and utilizes ECG technology to determine the HR can also be used. The PPG sensors common in many watch-based wearable devices have been shown to be much less accurate at reading HRs during exercise than ECG-based HR monitors, mainly due to the PPG sensor’s susceptibility to motion artifacts during movement [30,31,32,33,34]. ECG-based HR monitors have been recommended for use during exercise, which was observed in the current investigation. While WT represents an improvement in availability in tracking physiological metrics, such as the VO2max, field-based maximal and submaximal tests to estimate the VO2max have been around for decades [35]. A meta-analysis detailing the performance of these submaximal predictive equations compared to gold-standard testing found that they have a correlation range of r = 0.57 to 0.92 [36]. The current investigation found an r value of 0.78 for the 15 s and 30 s timeframes. Previous studies have found the Garmin fēnix 3 to have correlations of up to 0.92 [37], equal to the best submaximal equations that have been developed in terms of correlation values. Although comparing these devices solely based on correlation provides an imperfect view of their validity, accuracy, and reliability, they do offer some comparative value.

Having an accurate estimate of VO2max can be very useful, as it represents an important metric to determine a person’s health status. VO2max is a reliable predictor for overall cardiorespiratory fitness (CRF), which is an independent risk factor for all-cause and disease-specific mortality [14,15,16]; meaning, an individual with a low VO2max value will be at a higher risk of mortality due only to that metric, regardless of any other health metrics. The American Heart Association has released a lengthy review and position statement endorsing regular measurements of CRF in clinical practice. They concluded that a substantial body of epidemiological and clinical research indicates that cardiorespiratory fitness is a potent predictor of mortality, often surpassing the predictive power of established risk factors such as smoking, hypertension, hyperlipidemia, and diabetes mellitus. Incorporating CRF into risk stratification models can substantially enhance the precision of risk assessment for adverse health outcomes [38], as an assessment of CRF is ideally performed through a maximal exercise test and measurement of oxygen consumption and carbon dioxide production through a metabolic cart. Unfortunately, this is not possible for many people who cannot complete a maximal exercise test (those with CVD, musculoskeletal diseases, pulmonary diseases, etc.) or for those who cannot afford the cost of laboratory measurements. Wearable technology has the potential to evaluate a person’s VO2max through a relatively light bout of exercise (as is the case with the current device being tested) or even at rest (as is the case with other wearable devices). Thus, an accurate estimate of VO2max has the potential to influence personal health measures, as well as provide greater insights into the public health status for researchers and policymakers. As the fēnix 6 was found to generate accurate estimates of VO2max, individual recreational users, and possibly researchers, public health officials, and healthcare professionals can trust the values generated by the device. However, researchers and healthcare workers may want to utilize a more stringent validation threshold than what has been employed in the current investigation.

In addition to the role of VO2max in personal health, it is also an important measure for endurance athletes. VO2max is among the most important single measures to determine performance in an endurance event and is considered by many to be the single most important metric in determining performance [17,18,19]. Having the ability to know an athlete’s VO2max allows for improved training programs to be developed that are tailored to the athlete’s specific fitness level. As gold-standard methods of determining a person’s VO2max can be expensive and time-consuming, they are not a practical option for many recreational athletes or teams. Wearable technology can represent a cost-effective method of determining aerobic capacity for individuals, as well as teams. These devices can also generate a VO2max value during the course of normal training, eliminating the need to take a day off from training for testing purposes. It also has the added benefit of constant monitoring, allowing for small changes in aerobic capacity to influence the training protocol.

Measuring BOS via pulse oximetry is a well-established and widely used method in hospitals and other clinical settings. The introduction of pulse oximetry into smartwatches and other wearable devices is a recent advancement. Pulse oximeters measure BOS by broadcasting pulses of light and measuring the reflection via PPG sensors to monitor changes in blood oxygen concentration. This technology may prove to be an important way to monitor a person’s disease status and health metrics, especially those with pulmonary diseases, such as asthma, emphysema, and chronic obstructive pulmonary disease (COPD). However, independent validation of these devices will need to be completed in order to trust these measurements. It can also be useful for athletes who travel to altitude to monitor their acclimatization process, such as hikers, mountaineers, or other athletes traveling to higher altitudes than their current altitude [39]. While the device tested in the current investigation performed poorly, especially during the hypoxic conditions, it may be of interest to future researchers to test the ability of this technology to measure BOS levels accurately throughout the day rather than on demand. However, as we have mentioned previously, PPG sensors are susceptible to motion artifacts and could have similar issues with accuracy when measured throughout the day. Some research has demonstrated that desaturations below 50% can be observed when patients are moving during testing [40]. With these severe limitations in terms of the accuracy of this device, especially during hypoxic conditions, those looking to use this device to measure acclimatization when at altitude should look elsewhere for accurate measurements.

For this current investigation, we have used the generally accepted thresholds of a MAPE < 10% and CCC > 0.7. However, universal agreement for thresholds or even analytical tests to determine validity has not been established. As we recruited from the general population for this study, the fairly liberal thresholds of 10% and 0.7 seem appropriate. However, those looking to use this device in higher-level athletics, public health and/or physiology research, and healthcare may seek more conservative thresholds to determine appropriate use cases. In the future, a tiered threshold system could be established to better understand the appropriate use cases of these devices. In terms of the analytical tests, we have decided only to use MAPE and CCC in the determination of validity. However, we have also included bias assessments (Bland–Altman analysis) and equivalence testing (the TOST test). These have all been suggested as appropriate analytical techniques to determine validity, though they are not always common in other validation literature [13,41,42]. For instance, equivalence testing is especially absent from much of the validation literature. We have included all for the benefit of the reader as well as because we view them as appropriate tests to determine validity. However, because the thresholds have not been established for these additional tests, we have not included them in our validity thresholds.

### Limitations

This study evaluated both active and sedentary individuals in the general population, and thus the generalizability of this device to other populations should be done cautiously, if at all. While the validation criteria used have been used in previous research, the relatively liberal thresholds (MAPE < 10% and CCC > 0.7) might not be sufficiently stringent for high-stakes applications such as high-level athletics, public health research, or healthcare settings. As this study only evaluated acute hypoxia, additional research should be performed on these devices to determine their accuracy and usefulness in monitoring blood oxygen saturation longitudinally. Finally, we have noted the environmental conditions that VO2max was tested in, as it was an outdoor running trial. As the temperature can impact a person’s HR during exercise, this could be a confounding factor in the estimation. However, given that the data were collected over a ~14-month time period, this strengthens the external validity of the results and the generalizability.

## 5. Conclusions

In this study, we tested the Garmin fēnix 6 VO2max estimate and blood oxygen saturation values, measured via pulse oximetry for accuracy, and compared them to gold-standard laboratory measurements. The fēnix 6 showed acceptable accuracy for VO2max and was most closely aligned with the 15 s and 30 s timeframes. The fēnix 6 did not show acceptable accuracy for blood oxygen levels for any condition or the combined analysis. Therefore, the Garmin fēnix 6 may reasonably be expected to generate an accurate estimate of an individual’s VO2max based on 15 s or 30 s aggregated data if more accurate laboratory tests are not available. In addition, the fēnix 6 will not generate an accurate estimate of an individual’s blood oxygen levels, either in normoxia/hypoxia or utilizing anterior/posterior watch placement on the wrist.

## Figures and Tables

**Figure 1 sensors-25-00275-f001:**
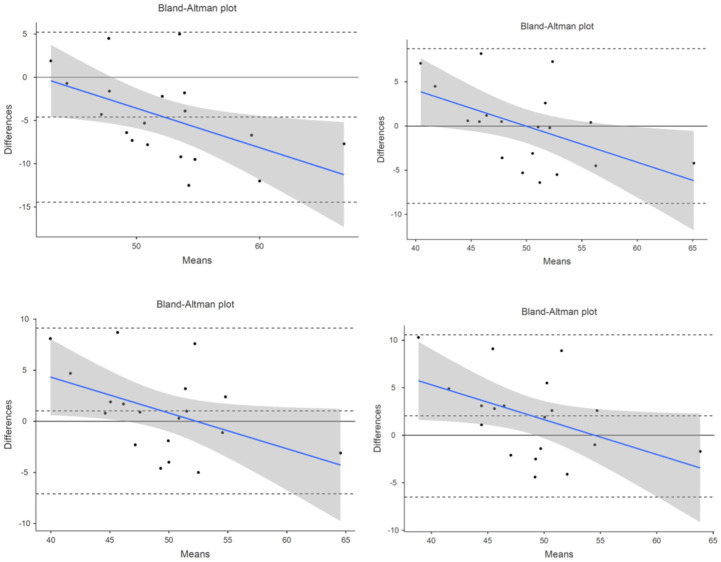
VO2 Bland–Altman plot of fēnix 6 compared to laboratory VO2max values: 4 s average in **top left**, 15 s average in **top right**, and 30 s average in **bottom left**, 1 min average in **bottom right**. Blue line represents proportional bias line with shadings representing 95% confidence intervals of proportional bias line. X-axis is the mean of the two measurements with the Y-axis the difference between the two measurements. The mean bias line and upper and lower limits of agreement are shown in dashed lines (mean bias being the middle-dashed line). The solid line represents the hypothetical mean bias of 0.

**Figure 2 sensors-25-00275-f002:**
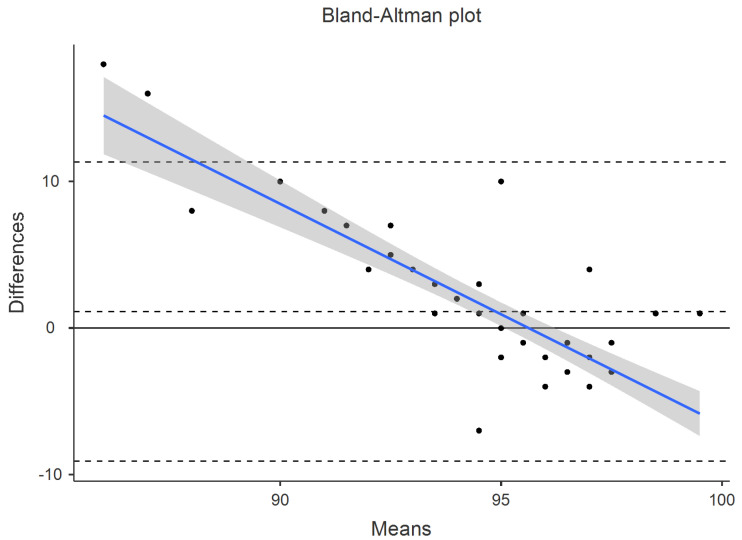
Bland–Altman plots for the combined pulse oximetry data, containing both hypoxia and normoxia conditions. Blue line represents proportional bias line with shadings representing 95% confidence intervals of proportional bias line. X-axis is the mean of the two measurements with the Y-axis the difference between the two measurements. The mean bias line and upper and lower limits of agreement are shown in dashed lines (mean bias being the middle-dashed line). The solid line represents the hypothetical mean bias of 0.

**Table 1 sensors-25-00275-t001:** VO2max descriptive and validation statistics results, n = 20. Notes: MAPE = mean absolute percentage error; TOST test = two one-sided *t*-tests. Bland–Altman bias values and 95% confidence intervals are provided. Values that met the predetermined validation criteria are bolded.

	Fēnix 6 VO2max Estimate	Lab VO2max—4 Breath Avg	Lab VO2max—15 s Avg	Lab VO2max—30 s Avg	Lab VO2max—1 min Avg
Mean (mL/kg/min)	49.68	54.54	49.95	48.94	47.91
Standard Deviation	4.61	7.28	7.04	6.67	6.76
MAPE		10.70%	**7.23%**	**7.05%**	**8.53%**
Pearson Correlation		0.73	0.78	0.78	0.76
Lin’s Concordance		0.49	**0.71**	**0.73**	0.68
Bland–Altman Bias		−4.87 (−7.30, −2.44)	−0.26 (−2.45, 1.92)	0.75 (−1.28, 2.78)	1.77 (−0.35, 3.89)
TOST Test (Upper)		<0.001	0.80	0.45	0.10
TOST Test (Lower)		<0.972	0.01	0.09	0.34

**Table 2 sensors-25-00275-t002:** Blood oxygen saturation measurements measured via pulse oximetry in Garmin fēnix 6 and criterion device. Descriptive and validation statistics results for n = 22 (52 distinct fēnix 6 values from all conditions and participants). Bland–Altman bias values and 95% confidence intervals are provided. Values that met the predetermined validation criteria are bolded.

	Fēnix 6 Blood Oxygen Saturation Measurement (%)	Criterion: Blood Oxygen Saturation Measurement (%)
Mean	95.44%	92.06%
Standard Deviation	1.60%	8.17%
MAPE		**4.29%**
Pearson Correlation		0.18
Lin’s Concordance		0.10
Bland–Altman Bias		1.12 (−0.34, 2.57)
TOST Test (Upper)		0.13
TOST Test (Lower)		0.02

## Data Availability

The data presented in this study are openly available in Excel format at https://doi.org/10.7910/DVN/43AX9B.

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
