# Peer review of "Validation of Aerobic Capacity (VO2max) and Pulse Oximetry in Wearable Technology"

_sensors, 2025, doi:10.3390/s25010275_

Round 1

Reviewer 1 Report

Comments and Suggestions for Authors

The study by Carrier et al. evaluated the accuracy of VO2max estimates and blood oxygen saturation measured via pulse oximetry using a wearable device (i.e. Garmin fēnix 6) in the general population. To this aim, two groups of subjects were considered, one including 19 subjects and the other including 22 subjects. In the following a list of minor and major comments.

Line 8. “the need for independent validation of these devices is needed”. The sentence needs rephrasing.

Lines 100-101. Apparently, there is a mismatch with the info reported at line 78 (19 participants).

Line 153-154. A reference is needed.

Line 67. A brief explanation about the wearable device should be included.

Line 102. There is a temperature variation of 12 degrees, which is very high. It is advisable to explain in which time frame the study was conducted and to discuss how such a high temperature differences may affect VO2max estimation.

Line 78 and Line 107 “apparently healthy” could be changed to: “people who, based on their personal knowledge, reported being healthy at the time the study was conducted”

Line 116 The authors should clarify what the “necessary steps” performed are.

Line 152. The meaning of “Each laboratory aggregated timeframe” should be detailed.

Lines 178-179. What are the four conditions the authors are referring to?

Line 199 “acceptable accuracy” should be quantified.

Line 244-249: I would avoid to cite a whole paragraph of another paper. I suggest to summarize the content of the paragraph with different words.

Author Response

Response to Reviewers Round 1: Validation of Aerobic Capacity (VO2max) and Pulse Oximetry in Wearable Technology

We would like to thank the reviewers for their time and thoughtful comments while reviewing this paper. It has surely improved the quality and readability of the paper. We have highlighted the changes in the newly uploaded document, as well as noted the changes below. We hope you approve.

The study by Carrier et al. evaluated the accuracy of VO2max estimates and blood oxygen saturation measured via pulse oximetry using a wearable device (i.e. Garmin fēnix 6) in the general population. To this aim, two groups of subjects were considered, one including 19 subjects and the other including 22 subjects. In the following a list of minor and major comments.

Line 8. “the need for independent validation of these devices is needed”. The sentence needs rephrasing.

  • Thank you, we have reworded the sentence at the reviewers request. It now reads “As wearable technology becomes increasingly popular and sophisticated, independent validation is needed to determine its accuracy and potential applications.”

Lines 100-101. Apparently, there is a mismatch with the info reported at line 78 (19 participants).

  • This typo has been fixed, good catch. It now reads “Five participants completed the testing at the track, and 14 participants completed testing on campus”.

Line 153-154. A reference is needed.

  • We kindly request some additional clarification. If the reviewer is requesting a citation regarding the thresholds, we do provide a paragraph discussing how there are not universally accepted thresholds, and these thresholds were simply what we chose, based on our judgement and what has been used previously in the literature (lines 298-314 of the original, and 303-319 of the updated manuscript). We cite the following in that paragraph:
    • Carrier, B.; Barrios, B.; Jolley, B. D.; Navalta, J. W. Validity and Reliability of Physiological Data in Applied Settings Measured by Wearable Technology: A Rapid Systematic Review. Technologies 2020, 8, 70.
    • Welk, G. J.; Bai, Y.; Lee, J.; Godino, J.; Saint-Maurice, P. F.; Carr, L. Standardizing analytic methods and reporting in activity monitor validation studies. Med. Sci. Sports Exerc. 2019, 51, 1767.
    • van Lier, H.,G.; Pieterse, M. E.; Garde, A.; Postel, M. G.; de Haan, H.,A.; Vollenbroek-Hutten, M.; Schraagen, J. M.; Noordzij, M. L. A standardized validity assessment protocol for physiological signals from wearable technology: Methodological underpinnings and an application to the E4 biosensor. Behavior research methods 2020, 52, 607–629.
  • Would the reviewer also like them to be cited in the methods section?

Line 67. A brief explanation about the wearable device should be included.

  • Thank you, we have added a brief description of the watch at the request of reviewer in the methods. It reads, “The Garmin fÄ“nix 6 is a rugged, multisport GPS smartwatch designed for outdoor use and athletes. It is marketed to combine the functionality of a fitness tracker, outdoor navigator, and smartwatch in a durable, wrist-worn device.”

Line 102. There is a temperature variation of 12 degrees, which is very high. It is advisable to explain in which time frame the study was conducted and to discuss how such a high temperature differences may affect VO2max estimation.

  • We have addressed this in two sections, the first is a sentence stating the timespan of data collection, and reads, “The data collection took place over the timespan of ~14 months, with running trials being completed during morning, afternoon, and evening.”
  • The second is in the newly added limitations section, which addresses the environmental concerns in the final part, and reads, “Finally, we have noted the environmental conditions that VO2max was tested, as it was an outdoor running trial. As temperature can impact HR during exercise, this could be a confounding factor in the estimation. Though as data was collected over a ~14 month time period, this strengthens the external validity of the results and the generalizability.”

Line 78 and Line 107 “apparently healthy” could be changed to: “people who, based on their personal knowledge, reported being healthy at the time the study was conducted”

  • Thank you, we have added this to the paper at the reviewers request. It now reads, “For VO2max testing, 19 apparently healthy (people who, based on their personal knowledge, reported being healthy at the time the study was conducted), active and sedentary individuals were recruited to participate”

Line 116 The authors should clarify what the “necessary steps” performed are.

  • Thank you, we have clarified this at the reviewers request. It now reads, “Researchers performed the necessary steps (selecting the correct icon in the watch) in the watch to generate a BOS level by the fÄ“nix 6 and recorded the value from the fingertip oximeter at the same time the watch generated a value.”

Line 152. The meaning of “Each laboratory aggregated timeframe” should be detailed.

  • Thank you, we have added this at the request of the reviewer. It now reads, “Data analysis for VO2max was completed by comparing the fÄ“nix 6 estimates of VO2max to each laboratory aggregated timeframe (4-breath, 15-sec, 30-sec, 1-min).”

Lines 178-179. What are the four conditions the authors are referring to?

  • We clarified the conditions at the request of the reviewer. It now reads, “Error analysis showed that the fÄ“nix 6 BOS values had a MAPE of less than 10% for all four conditions and the combined data (anterior/posterior, hypoxia/normoxia; see Table 2 and supplementary files).”

Line 199 “acceptable accuracy” should be quantified.

  • Thank you, we have clarified this at the request of the reviewer. It now reads, “In this study, the validity of VO2max estimates and blood oxygen saturation (BOS) values measured via pulse oximetry in wearable technology (WT) was compared to gold standard measurements. Based on the pre-established validation criteria, the fÄ“nix 6 has acceptable accuracy (MAPE < 10%, CCC > 0.7) in its estimation of VO2max and corresponds closely to the 15-sec and 30-sec averaged timeframes.”

Line 244-249: I would avoid to cite a whole paragraph of another paper. I suggest to summarize the content of the paragraph with different words.

Thank you, we have updated this to paraphrase, rather than quote, at the request of the reviewer. It now reads, “They concluded that a substantial body of epidemiological and clinical research indi-cates that cardiorespiratory fitness is a potent predictor of mortality, often surpassing the predictive power of established risk factors such as smoking, hypertension, hyper-lipidemia, and diabetes mellitus.”

Reviewer 2 Report

Comments and Suggestions for Authors

It was with enthusiasm that I volunteered to review this manuscript. This manuscript sought to evaluate the efficacy of a wearable device on measures of VO2 and pulse ox compared to known standards, under two different conditions. The strength of this manuscript was the methodology. It allowed the researchers to ask, and quantify, a very specific question. And the authors are totally correct, these devices keep doing more and more, are not regulated, and no one knows if what the device says it does is correct. So...kudos to the authors for taking the initiative and asking a very pertinent question. I do have some random thought that I think can help the reader.

Abstract:

-On line 27 the authors use 'WT". That has not been presented as of yet. 

Intro:

-Page 2, line 57-59. The authors reference three manuscripts stating that VO2 is most important when describing performance. However, for one of these that is not true. Reference 18., Bassett and Howley, the abstract states, 'The speed at lactate threshold (LT) integrates all three of these variables and is the best physiological predictor of distance running performance.' 

Methods:

-Page 2, line 79, Page 3, line 108. Both times that the authors present BMI there are no units.

-Page 3, line 100-101. The authors state that five participants completed track testing, and 15 participants completed testing on campus. If my math is correct that is 20 participants. Everywhere in the manuscript it says 19 participants. 

-Page 3, line 100. Is five minutes enough time in hypoxia to adapt? Do the authors have a reference for that? 

My other random comment is about the random lines that are in bold. It makes reading the document challenging to some degree. 

Author Response

Response to Reviewers Round 1: Validation of Aerobic Capacity (VO2max) and Pulse Oximetry in Wearable Technology

We would like to thank the reviewers for their time and thoughtful comments while reviewing this paper. It has surely improved the quality and readability of the paper. We have highlighted the changes in the newly uploaded document, as well as noted the changes below. We hope you approve.

Reviewer 2:

It was with enthusiasm that I volunteered to review this manuscript. This manuscript sought to evaluate the efficacy of a wearable device on measures of VO2 and pulse ox compared to known standards, under two different conditions. The strength of this manuscript was the methodology. It allowed the researchers to ask, and quantify, a very specific question. And the authors are totally correct, these devices keep doing more and more, are not regulated, and no one knows if what the device says it does is correct. So...kudos to the authors for taking the initiative and asking a very pertinent question. I do have some random thought that I think can help the reader.

  • We would like to thank the reviewer again for the time and thoughtful review of this paper (and for the kudos).

Abstract:

-On line 27 the authors use 'WT". That has not been presented as of yet. 

  • Thank you, this has been corrected.

Intro:

-Page 2, line 57-59. The authors reference three manuscripts stating that VO2 is most important when describing performance. However, for one of these that is not true. Reference 18., Bassett and Howley, the abstract states, 'The speed at lactate threshold (LT) integrates all three of these variables and is the best physiological predictor of distance running performance.' 

  • Good catch. We have updated the language to better reflect the nuance. It now reads, “VO2max also has an important relationship to endurance performance among athletes, often being cited as the most important single factor - or among the most important factors - in predicting race performance (17-19)”

Methods:

-Page 2, line 79, Page 3, line 108. Both times that the authors present BMI there are no units.

  • Thank you, as BMI is an index, it is not always expected to report the units. However, many times they are included, and we have added them in at the request of the reviewer.

-Page 3, line 100-101. The authors state that five participants completed track testing, and 15 participants completed testing on campus. If my math is correct that is 20 participants. Everywhere in the manuscript it says 19 participants. 

  • Good catch, the other reviewer noted this and it has been updated.

-Page 3, line 100. Is five minutes enough time in hypoxia to adapt? Do the authors have a reference for that? 

  • 5 minutes was long enough for them to stabilize, as the cardiovascular system is a quickly adapting system, then testing was done beyond that. Though steady state would be a factor of the level of hypoxia, and while ours was high altitude, it was still within normal levels for human life, and therefore would have a relatively quick adaptation period. However, we have added a limitations section that addresses the limitation of acute monitoring vs longitudinal monitoring of BOS. Additionally, as testing was done simultaneously, we do not require them to be at the same BOS as another trial, as the criterion and test devices are monitoring the BOS in real-time. And while a reference stating that 5 minutes is long enough to adapt is more difficult to find, as the length of time in hypoxia impacts the body and different systems differently, the following references may be of interest to the reviewer.
    • Ursino, M., & Magosso, E. (2000). Acute cardiovascular response to isocapnic hypoxia. I. A mathematical model. American Journal of Physiology-Heart and Circulatory Physiology, 279(1), H149-H165.
    • Peltonen, J. E., Tikkanen, H. O., & Rusko, H. K. (2001). Cardiorespiratory responses to exercise in acute hypoxia, hyperoxia and normoxia. European journal of applied physiology, 85(1), 82-88.
    • Williams, A. M., Levine, B. D., & Stembridge, M. (2022). A change of heart: Mechanisms of cardiac adaptation to acute and chronic hypoxia. The Journal of physiology, 600(18), 4089-4104.

My other random comment is about the random lines that are in bold. It makes reading the document challenging to some degree. 

This may have been an issue with the system converting it to whatever file type you downloaded it as. As the only bolded lines we see are set with the Sensors template. Anyways, we assume any formatting issues will be addressed by the editors when we go to print. We will watch out for any unnecessarily bolded lines. Thank you.

Round 2

Reviewer 1 Report

Comments and Suggestions for Authors

The authors solved all the raised issues.